# Advances in the Lung Cancer Immunotherapy Approaches

**DOI:** 10.3390/vaccines10111963

**Published:** 2022-11-19

**Authors:** Hafiza Padinharayil, Reema Rose Alappat, Liji Maria Joy, Kavya V. Anilkumar, Cornelia M. Wilson, Alex George, Abilash Valsala Gopalakrishnan, Harishkumar Madhyastha, Thiyagarajan Ramesh, Ezhaveni Sathiyamoorthi, Jintae Lee, Raja Ganesan

**Affiliations:** 1Jubilee Centre for Medical Research, Jubilee Mission Medical College and Research Institute, Thrissur 680005, Kerala, India; 2Life Sciences Industry Liaison Lab, School of Psychology and Life Sciences, Canterbury Christ Church University, Sandwich CT13 9ND, UK; 3Department of Biomedical Sciences, School of Biosciences and Technology, Vellore Institute of Technology (VIT), Vellore 632014, Tamil Nadu, India; 4Department of Cardiovascular Physiology, Faculty of Medicine, University of Miyazaki, Miyazaki 889-1692, Japan; 5Department of Basic Medical Sciences, College of Medicine, Prince Sattam bin Abdulaziz University, P.O. Box 173, Al-Kharj 11942, Saudi Arabia; 6School of Chemical Engineering, Yeungnam University, Gyeongsan 38541, Republic of Korea; 7Institute for Liver and Digestive Diseases, College of Medicine, Hallym University, Chuncheon 24253, Republic of Korea

**Keywords:** lung cancer, immunotherapy, epidemiology, immune profiling, vaccines: combinatorial therapy, cancer models

## Abstract

Despite the progress in the comprehension of LC progression, risk, immunologic control, and treatment choices, it is still the primary cause of cancer-related death. LC cells possess a very low and heterogeneous antigenicity, which allows them to passively evade the anticancer defense of the immune system by educating cytotoxic lymphocytes (CTLs), tumor-infiltrating lymphocytes (TILs), regulatory T cells (Treg), immune checkpoint inhibitors (ICIs), and myeloid-derived suppressor cells (MDSCs). Though ICIs are an important candidate in first-line therapy, consolidation therapy, adjuvant therapy, and other combination therapies involving traditional therapies, the need for new predictive immunotherapy biomarkers remains. Furthermore, ICI-induced resistance after an initial response makes it vital to seek and exploit new targets to benefit greatly from immunotherapy. As ICIs, tumor mutation burden (TMB), and microsatellite instability (MSI) are not ideal LC predictive markers, a multi-parameter analysis of the immune system considering tumor, stroma, and beyond can be the future-oriented predictive marker. The optimal patient selection with a proper adjuvant agent in immunotherapy approaches needs to be still revised. Here, we summarize advances in LC immunotherapy approaches with their clinical and preclinical trials considering cancer models and vaccines and the potential of employing immunology to predict immunotherapy effectiveness in cancer patients and address the viewpoints on future directions. We conclude that the field of lung cancer therapeutics can benefit from the use of combination strategies but with comprehension of their limitations and improvements.

## 1. Introduction

Lung cancer (LC) is the primary reason for cancer-related death worldwide. Over the previous ten years, progress in overall survival (OS) has been reported as a result of new, efficient medicines that have developed in part due to advancements in the management of non-small cell lung cancer (NSCLC) [1]. Numerous authorized targeted drugs are the therapy of choice for NSCLC patients with genetic alterations [2]. NSCLC is caused by a biologically complex collection of several oncogenes. A downside of targeted therapy is the inability to identify suitable oncogenes to target cancer cells. Immune checkpoint inhibitors (ICIs), the current leading candidate in immunotherapy, are currently a vital component for NSCLC therapy. Biomarkers that can indicate how well a patient will respond to checkpoint inhibitors include Programmed cell death ligand-1 (PD-L1) and CTLA-4-cytotoxic T-lymphocyte antigen 4 (CTLA-4). Although PD-L1 expression is not ideal, it is currently the most reliable clinical biomarker. Patients in advanced LC stages exhibit disease regression with first-line treatment, as well as the second-line choices are restricted in patients without a targetable gene mutation [3]. Additionally, the clinical utility of ICIs in the administration of small cell lung cancer (SCLC) is competitively less than that of NSCLC [4]. It is still imperative to increase immune system activation, broaden the range of available therapies, and prevent ICI resistance to improve the clinical efficacy of LC [5]. 

In this review, we discuss the epidemiologic trend of LC immunotherapy approaches using data curated from The Cancer Genome Atlas (TCGA)-cBioportal. We also focus on the current trends and advances of LC immunotherapy, importantly LC-immunoprofiling, preclinical and clinical studies in NSCLC and SCLC considering cancer models and vaccines, and also the emerging combinatorial approaches using immunotherapy. 

## 2. Current Lung Cancer Epidemiology

Substantial changes have been experienced in LC epidemiology, considering incidence, prevalence, and mortality during the past several decades [6]. The WHO mortality predictions and the epidemiologic trends of different malignancies reported on the Global Cancer Observatory (GLOBOCAN)-2020 revision are in agreement [7]. The frequent occurrence, prevalence, and mortality rates of LC highlight the need for efficient therapeutic approaches. Cancer in its early-stage identification is vital because the localized stage has a competitive benefit in terms of relative five-year survival over regional and distant stages. Over NSCLC, which makes up around 85% of all LC types, SCLC is the most prevalent epithelial LCs [8]. According to Surveillance, Epidemiology and End Result program (SEER), distant stage LC has a greater diagnosis rate (56%) and a lower relative five-year survival rate (6.3%) than regional and local stages [9]. About 15% of all cases of LCs are SCLC, in which most patients (60–70%) have advanced stages of their diagnoses. Less than 10% of patients with advanced illnesses survive for five years [8]. We have curated TCGA-cBioportal data with the keywords LC, clinical prognosis, and immunotherapy and have plotted them in Figure 1. The OS rate of LC patients is poor and is associated with the stage of LC initially diagnosed (Figure 1a). The PD-L1 expression in various tissue sites (Figure 1d) of LC cases has given a higher frequency for lymph nodes followed by bone, pleura, liver, brain, and pleura, using the protein expression data from 31 studies (Figure 1c), and various histotypes (Figure 1b). We have compared the number of cases that have undergone radiation therapy, chemotherapy, targeted therapy, and immunotherapy, and it is higher in chemotherapy (683/175) and lower in radiation therapy (158/822) followed by immunotherapy (191/666) (Table 1) [10]. The data represent that only 28.6% of LC patients underwent immunotherapy.

## 3. Five Pillars of Lung Cancer Therapy

Traditional cancer treatment approaches can be divided into five main categories: surgery, chemotherapy, radiation therapy (also known as external radionuclide therapy (ERT)), targeted therapy, and immunotherapy which has recently been introduced as a fifth category [11]. Clinical cancer care has undergone a profound transformation with specific flaws in each of these pillars that have now been targeted by designer customized medicines intended to enhance survival rates and lessen side effects. The entire lung tumor, as well as any adjacent lymph nodes, are to be removed during surgery [12]. Radiation therapy can be used to treat LC in its early stages, but like surgery, it cannot be applied in invasive cases. The cancer cells that are in the exposure to radiation are killed by radiation therapy. The healthy cells in its path are likewise harmed [13]. Neoadjuvant and adjuvant therapies are preferred due to disease recurrence in LC patients, respectively, before and after the surgery [7]. Chemotherapy, targeted therapy, and immunotherapy fall into adjuvant therapy subtypes wherein medications kill cancer cells (Table 2) [14]. Cancer cells are prevented from proliferating, dividing, and producing new cells by chemotherapy [15]. It has been demonstrated to enhance OS in patients with LC at all stages. A chemotherapy regimen normally comprises a specific number of cycles delivered over a predefined duration of time. The medications suggested for chemotherapy depends on cancer histotypes, such as adenocarcinoma (ADC) or squamous cell carcinoma (SQCC). The chemotherapy side effects vary depending on the dosage and the patient. A treatment known as targeted therapy specifically targets the unique proteins, genes, or tissue environment that promotes the survival of the disease. This form of therapy prevents the proliferation and invasion of cancer cells while minimizing harm to healthy cells [16]. Immunotherapy, often known as biological therapy, aims to strengthen the inherent defenses against cancer. It uses materials that can be produced in vivo or in vitro to restore immune system function. Immunotherapy for NSCLC patients may be administered as a single medication, in combination with other immunotherapy medications, or in conjunction with chemotherapy. Immunotherapy or combinatorial approaches can be considered in cases where targeted therapy cannot be applied [17]. For instance, immunotherapy outperformed chemotherapy in NSCLC patients with high expression of PD-L1. For example, Phase III trial KEYNOTE-042 trial compared the efficacy of immunotherapy (pembrolizumab) over chemotherapy (platinum-based) in 1274 patients with PD-L1 expression of greater than 1% Tumor Percentage Score (TPS). Pembrolizumab had better OS comparing chemotherapy (16.7 months, 12.1 months, respectively), but the PFS was more significant for chemotherapy (5.4 months, 6.5 months, respectively) [18]. 

## 4. Lung Cancer Agonistic and Antagonistic Immune Cells 

Although LC immunosurveillance may be successful in the early stages of oncogenesis, it is hindered when a clinically detectable tumor has formed. LC cells have a very low and heterogeneous antigenicity, allowing them to passively evade the anti-cancer defense of the immune system. Mesenchymal, epithelial, and endothelial cells, in addition to a substantial number of immune cells tangled in a sophisticated cytokine network, serve the lung’s immune system. The cytokines and the cellular components in the immune system interact and are stably integrated into healthy individuals [45] but are distorted in cancer conditions [46,47]. The antitumor response requires two major immune cells: non-specific cells and cytotoxic T lymphocytes (CTLs). Non-specific cells can recognize and present the tumor cell antigen to CTLs [45,48,49]. The malignant tissue should be invaded by activated T lymphocytes that can identify the neoplastic antigens. The intracellular cytotoxic proteins from CTLs are released at the site of neoantigens and in combination with general mechanisms offered by natural killer (NK) cells or macrophages. The interactions between the tumor microenvironment (TME) and the immune system are obviously diverse and dynamic, impacting carcinogenesis [45,50,51]. LC agonistic and antagonistic immune cells in terms of ICI effectiveness and immune activation can be explained with (a) tumor with an immunological exclusion, where the immune cells are limited to the stroma or the margin of the tumor; (b) cold tumors, where a significant inflammatory response is not present in tumor cells; (c) hot tumors, in which T lymphocytes are heavily infiltrated, and various inflammatory signals are activated [52,53,54]. 

Immunological cells that lack the ability to homing to the tumor bed are another definition of tumors with immune exclusion [55]. Angiogenesis, elevated transforming growth factor (TGF)-β signaling, and myeloid polarization are all characteristics of immune-excluded tumors [55,56]. Additionally, T cell distribution at the boundary of the tumor is also a characteristic. Pai et al. hypothesized different mechanisms most likely in charge of immunological exclusion. There are physical barriers that stop T cells from encountering cancer cells directly as an initial anti-tumor response [55]. The primary contributing factors include increased Vascular Endothelial Growth Factor (VEGF) secretion and their cross-talks [55,57]. Second, there might be functional barriers made up of biochemical and metabolic cross-talk between immune, stromal, and cancer cells [58,59]. The deregulation of Wnt/β-catenin signaling and phosphatase and tensin homolog (PTEN) leads to the inhibition of CTL from invading tumor tissue [55,59,60]. Additionally, TME is enriched with metabolites, including indoleamine 2,3-Dioxygenase (IDO), nitric oxide (NO), and arginase [61].

Cold tumors are characterized by the absence of tumor-inducing immune cells inside the tumor tissue [52]. In addition, large amounts of pro-inflammatory cytokines or other metabolic factors, such as NO, IDO, and arginase, are not seen in the microenvironment of this kind of neoplasm. A single-driver mutation is thought to be more prevalent in cold tumors because neither a significant TMB nor the presence of neoantigens are seen in these tumors [62]. However, Treg cells and MDSC may be present in the TME around cold tumors [63,64,65,66]. A low concentration of adhesion molecules such as CD34, intercellular adhesion molecule (ICAM), vascular cell adhesion molecule (VCAM), and E-selectin, together with MDSC and Treg cells, stimulate angiogenesis and metastasis. MDSC and Treg cells also inhibit the tumor-suppressing activity of dendritic cells [52,63,67]. 

Hot tumors are characterized by significant inflammatory signals inside the tissue, both pro-inflammatory cytokines and inflammatory cell infiltration [54,68]. However, the immune reaction in this form of cancer is incredibly ineffectual even though the tumor cells are very heavily infiltrated by tumor-suppressing immune cells [53,68]. Tumor-associated macrophages (TAMs) with their pro-tumor characteristics are strongly infiltrated in hot tumors along with non-specific immune cells [69,70]. TGF-β, IL-10, and other anti-inflammatory cytokines secreted by M2 macrophages have immunosuppressive effects in TME. T cell and NK cell anti-tumor activities are less stimulated because of the inefficiency of TAM to present tumor-associated antigens [63,69,70]. In particular, interferon (IFN)-γ, which mediates PD-L1 expression on tumor cells will be masked in lymphocytes [64,65]. 

Macrophages, lymphocytes, and granulocytes are the major antagonistic candidates in cancer progression. TAM [71,72], tumor-infiltrating lymphocytes (TIL) [73,74], tumor-associated tissue eosinophilia (TATE) [75], and tumor-associated neutrophils (TAN) [76] are the acronyms for the domains of macrophages, lymphocytes, eosinophils, and neutrophils respectively [75]. CTLs are the predominant cell population with activity in the immune response against cancer [77]. Natural killer T cells (NKT), CD8+ and CD4+ lymphocytes, and B lymphocytes make up the lymphocyte population [78,79]. Cancer cells are often destroyed by the cytolytic reaction or by apoptosis. Tumor cells and antigen-presenting cells (APC) must effectively offer antigens for the cytotoxic onslaught to be successful. Dendritic Cells (DCs) and macrophages play a major role in doing this [80]. After coming into touch with cancer antigens, APC delivers neoantigens to activate effector cells. Co-stimulatory molecules on APC and associated receptors on lymphocytes serve a critical role in the propagation of APC-lymphocyte signaling [77,81]. As major candidates in the CTL candidates, cytotoxic CD4+ lymphocytes, and CD8+ cells, the signal pathway B7-CD28 is extensively studied [77]. APC-CTL interaction was found to be blocked in malignancies. It is generally known that tumor inhibitory response is inadequate in larger solid tumor masses [82]. Low costimulatory molecule expression and low antigen presentation help LC cells conceal themselves from the cytotoxic onslaught. The various epigenetic and genetic changes during oncogenesis also result in the instability and poor definition of the LC antigen. Together, these factors cause cancer cells to passively evade immunosurveillance. However, many other aspects of this escape involve the intentional control and inhibition of the anti-cancer response of the immune system. Interactions between PD-1/PD-L1, FAS/FASL, and the secreted cytokines, including TGF-β, IL-6, and IL-10, induce CTL inhibition [83]. Cytotoxic T cells (Tc), Treg, Helper T cells (Th), NK cells, and B lymphocytes express PD-1. High levels of the PD1 ligands B7-H1 (CD274) and PD-L2 (CD273, B7-DC) are expressed by immune cells in TME. The interaction between PD1/PD-L1 and FAS/FASL has potent immunosuppressive properties [84,85,86]. Along with a higher percentage of FAS-positive lymphocytes and CTLs expressing a lot of the FAS receptor, cancer cells are also known to express a lot of the FAS-L protein. NSCLC has also been associated with changes in the amount of the soluble forms of FASL and FAS, which downstream regulate apoptosis. As a result, this receptor pathway is crucial to CTL number reduction [87]. There is presently no treatment being tested in a clinical setting that targets this pathway. Co-stimulatory molecules on APC and cancer cells can get modified as a mechanism for reducing anticancer resistance and masking cancer cells from CTL attack [88]. 

The CD28 receptor on lymphocytes (co-stimulatory molecule) and the B7 molecule (CD80/CD86) on the APC are required to initiate the cytotoxic action. B7 molecules can, however, also deliver a suppressive effect when they are linked to CTLA-4 [89,90,91]. CTLA-4 can block the T cell receptor (TCR) through sharing homology with CD28. By inhibiting CD28, CTLA-4 prevents the progression of the cell cycle and reduces IL-2 secretion while increasing the secretion of TGF-β. CTLA-4 expressed on Treg cells can induce cancer regulatory effects via interaction with forkhead box P3 (FOXP3) [92,93,94]. There are two ways that CTLA-4 is expressed: intracellularly as storage and on the cell surface following activation. T cells express CTLA-4 differently in LC than in healthy individuals. The intracellular domain of CTLA-4 is altered in cancer patients, while the surface expression is noticeably higher in cancer patients [95,96]. 

## 5. Immunophenotyping of Lung Cancer

A favorable correlation exists between the cancer prognosis and the intensity of immune cell infiltration into neoplastic tissue [97,98,99]. LC patient prognosis may be significantly influenced by the knowledge about the immune system state in tumor tissue [100,101,102]. CD8+ cell infiltration, non-Treg infiltration, IL12Rβ2 expression, CD69 expression on T cells, and the granzyme secretion contribute to a better prognosis, while neutrophil infiltration, IL-7R expression, and Treg cell infiltration represent worse prognosis. ICI therapy efficacy is in association with infiltration of CD8+, CD3+, CD19+ T cells, CD68+ macrophages, and PD-L1 expression [103]. Rather than the cytotoxic actions of T lymphocytes, they may be functionally inhibited in TME (for instance, tumor cell mutations in the janus kinase (*JAK*)*1* and *JAK2* genes may result in incorrect antigen presentation) [66,104,105]. The presence of TIL is a positive prognostic indicator in LC [106,107,108,109]. The presence of a high density of FOXP3+ lymphocytes and a higher proportion of stromal FOXP3+ cells to CD3+ cells is a reliable indication of stage II ADC recurrence [109]. Additionally, high expression of IL-12R is linked to a better outcome for patients with early-stage IA and IB, whereas IL-7R expression is a significant marker for the poor OS. Patients might even be classed for immunotherapeutic approaches focusing on the biology of TME [109]. Immunohistochemical analysis of TIM and TIL infiltration and c-Kit+ mast cells in cancer stroma in advanced ADC cases who received chemotherapy was performed. Patients who had much higher levels of TIM and TIL found in cancer tissue had a significantly better prognosis than those whose infiltration was predominately stromal. However, the quantity of immune cells in either tumor clusters or stroma and treatment response were not significantly correlated. On the other hand, Kinoshita et al. showed that CD8+ T cells are classified as a poor predictive marker in the TME of non-smoking LC patients. These cells expressed numerous immunoregulation genes at high levels and were immunodysfunctional in phenotype. On the other hand, high levels of IFNγ and granzyme-secreting activated CD8+ T cells were associated with postoperative survival in those patients [107].

### Impact of Immune Profiling and Scoring on Lung Cancer Prognosis

More PD-1 and CD8+ T cells were identified within the tumor and tumor borders of samples from pembrolizumab-treated patients [110,111], which indicates the predictive value of ICIs in cancer patients. Immunophenotyping of cancer cells is not an ICI predictor in prospective trials, including LC patients receiving immunotherapy [112,113]. The POPLAR research is one of the few retrospective experiments that correspond to immune system functional studies in tumor tissues. The effectiveness of atezolizumab and docetaxel in treating patients with locally progressed or metastatic NSCLC was compared. Immune-related gene signature profiles by using NGS were analyzed for the mechanisms corresponding to immune cell activation, like IFNγ signaling and immune cytolytic activity, and the results were compared. The genes linked to T-effector cell activation (*GZMB*, *CD8A*, *IFNγ*, *CXCL10*, *GZMA*, *CXCL9*, *TBX2,* and *EOMES*) was associated significantly with therapeutic significance in individuals who underwent anti-PD-L1 immunotherapy. Anti-PD1 inhibition is more effective against hot tumors, which are linked to higher gene expression for pro-inflammatory markers. The relationship between immunotherapy effectiveness and the immune environment in LC tissue was shown by Hwang et al. [114,115]. Using an Oncomine Immune Response Research Assay, 395 immune-related genes were examined in the pretreatment tumor sample. Patients with advanced NSCLC availing of anti-PD1 immunotherapy were classified into a non-durable clinical benefit (NDCB) and durable clinical benefit (DCB). DCB greatly outperformed NDCB in terms of the percentage of lymphocytes that infiltrated the center of the tumor. Gene profiles of peripheral T cells and M1 macrophages had the greatest influence on discriminating between NDCB and DCB patients. Genes with strong expression in the group M1 include *CD48*, c-c motif chemokine receptor 7 (*CCR7*), *CD27*, major histocompatibility complex class I G *(HLA-G*)*,* forkhead box O1 *(FOXO1)*, *HLA-B,* lysosomal associated membrane protein 3 (*LAMP3*), and NFκB inhibitor alpha *(NFKBIA)*, whereas genes with high expression in the peripheral T cell signature include *HLA-DOA*, G protein-coupled receptor 18 (*GPR18*), and signal transducer and activator of transcription 1 (*STAT1*). Additionally, the authors discovered that the highest expression of proteasome subunit beta type-9 (*PSMB9*) and *CD137* among the several examined genes were indicative of a long-lasting therapeutic benefit from anti-PD1 immunotherapy. Hwang et al. amply show the excessive complexity of the interactions necessary for an efficient immune response in neoplastic disorders. It signifies the accuracy of multigene analysis over TMB status evaluation or PD-L1 expression alone. Moreover, this study demonstrated the need for major efforts to be made to gather accurate information regarding the specific (peripheral T cell) and non-specific (M1) signatures of the immune response [115]. Automated image analysis, as carried out by Althammer et al., is a new method to identify immunological predictive markers, such as the presence of CD8+ T cells and PD-L1 expression. Anti-PDL1 expressing NSCLC cases who received immunotherapy were digitally graded for the density of CD8- and PD-L1 + ve cells. The median OS for durvalumab-treated PD-L1 and CD8 double +ve tumors and CD8 and PD-L1 negative tumors were 21 and 7.8 months, respectively. Only in ICI-treated patients, PD-L1-and CD8- double positive signatures offered a more accurate classification of OS than single high levels of PD-L1 or CD8+ cells. A single dense population of CD8+ cells, however, was substantially related to longer median OS (67 months) for immunotherapy-naive patients compared to the group with reduced CD8+ cell concentration [116]. 

## 6. Immunotherapy-Based Clinical Studies in Lung Cancer

Immunotherapy, particularly ICIs, has replaced chemotherapy as the primary treatment due to its improved survival rates and manageable adverse effects. Moreover, immunotherapy has the potential for better OS comparing other LC therapies. Both PD-1 and PD-L1 inhibitors are now considered a crucial component of managing unresectable and locally advanced LC treatment options [81]. In this section, we will discuss major NSCLC and SCLC clinical trials corresponding to immunotherapy.

### 6.1. Immunotherapy in Non-Small Cell Lung Cancer

The therapeutic potential of nivolumab (PD-1 inhibitor) was examined in the CA209-003 Phase I study in multiple cancer types, which consisted of 122 patients with NSCLC, and has resulted in RR of 17%, 5-year OS rate of 16%, and a median response of 17 months [117]. KEYNOTE001, a phase I investigation of pembrolizumab, was analyzed in 495 NSCLC patients, which involved immunohistochemical detection of PD-L1 using a 22C3 clone [118]. It resulted in an RR of 19.4%. Moreover, 53 patients with NSCLC were considered in the atezolizumab (targets PD-L1) phase I study, which revealed an RR of 23% [119]. Another PD-L1 clone, SP142, was used to measure PD-L1 expression, however, the outcomes were consistent. Randomly selected phase III trials (CheckMate017) were used to test the effectiveness of three ICIs against the docetaxel-based standard second-line treatment. Docetaxel 75 mg or Nivolumab 3 mg monotherapy was the second-line treatment option for 272 patients with squamous NSCLC [120]. The study showed significant OS and outperformed second-line docetaxel in almost all significant outcomes. Nivolumab had an RR of 20% compared to docetaxel’s RR of 9%. Nivolumab was related to a longer median PFS of 3.5 months and reduced toxicity.

In the OAK study, 850 NSCLC patients with any histology were randomized to receive either 1200 mg of atezolizumab monotherapy or 75 mg of docetaxel every three weeks. Atezolizumab provided better chances of survival. Atezolizumab and docetaxel showed a median OS of 13.8 months and 9.6 months, respectively. The OS of non-squamous and squamous LC had similar outcomes across histologic subtypes. Additionally, atezolizumab showed OS benefits in all PD-L1 strata, including those with low, moderate, high, and unfavorable outcomes. Atezolizumab (15%) had a lower incidence of grades 3–4 TRAEs than conventional docetaxel (43%) [121].

In individuals with advanced NSCLC, anti-PD-L1 treatments do not work similarly. The JAVELIN Lung 200 study compared the anti-PD-L1 antibody avelumab to docetaxel [122]. In this phase III randomized trial, 792 NSCLC patients’ recurrence after receiving platinum-doublet chemotherapy and with a minimum of 1% of cells expressing PD-L1 were enrolled. Despite having fewer Grade 3 or higher TRAEs than docetaxel (10% vs. 49%), a significant OS was not achieved. Avelumab and docetaxel had a median OS of 11.4 and 10.3 months, respectively.

Ipilimumab (anti-CTLA-4 antibody) is a typical candidate for renal cell carcinoma (RCC) and melanoma therapy. CheckMate 227 study shows the effectiveness of CTLA-4 in immunotherapy. It is a common first-line choice in NSCLC. In the MYSTIC study, 1118 NSCLC cases with previously untreated EGFR and ALK wild-type NSCLC were categorized into treatment: chemotherapy alone, durvalumab, durvalumab plus four doses of tremelimumab, or durvalumab plus tremelimumab [123]. Although there was no PD-L1 selection for study enrollment, the results were analyzed in 488 NSCLC cases with a minimum of 25% PD-L1 expression. There was a numerical but not statistically relevant increase in survival with immunotherapy in the primary efficacy analysis cohort. Compared to 12.9 months with chemotherapy, the median OS was 16.3 months with durvalumab. Durvalumab with tremelimumab resulted in a median OS of 11.9 months. Durvalumab, durvalumab plus tremelimumab, and chemotherapy had median PFS of 4.6, 3.9, and 5.4 months, respectively. 

### 6.2. Immunotherapy in Small Cell Lung Cancer

The goal of immunotherapy and their combinatorial approaches involving chemotherapy is to maximize the efficiency of the immune system fight cancer by fostering a favorable environment [124]. In the phase 2 STIMULI study, which included 153 limited-stage small cell lung cancer (LS- SCLC) cases, the treatment with nivolumab and ipilimumab showed a one-year OS rate of 79%, while 89% in the observational group. The 3-year OS was 54% in the consolidation group but 41% in the other arm, with no significant difference in PFS in both arms [125]. 

The phase 3 CASPIAN trial demonstrated superior survival results in durvalumab-platinum etoposide (PE) (another PD-L1 inhibitor) treated patient groups [126]. In this trial, patients in the control arm were compared to the test arm who received 4–6 rounds of durvalumab plus tremelimumab, carboplatin, or cisplatin in combination with etoposide and durvalumab. In comparison to the conventional chemotherapy arm, durvalumab contributed to OS of 2.4 months. Contrary to IMPOWER 133 study, durvalumab raised RR by 10% (67.9% in the test arm vs. 58% control arm) and significantly favored immunotherapy in all populations with or without brain metastases. Tremelimumab in combination with durvalumab and chemotherapy, and these results indicated a trend toward increased survival.

The phase 2 ECOGACRIN EA5161 research used nivolumab as the first-line therapy for ES-SCLC. Every three weeks, nivolumab was used with four cycles of chemotherapy. In responding patients, nivolumab 240 mg was given in every two weeks. In terms of OS (11.3 vs. 8.5 months) and PFS (5.5 vs. 4.7 months), nivolumab showed statistically significant improvement. Moreover, the nivolumab arm with no significant toxicities was noted [127]. 

The efficiency of nivolumab alone and in combination with ipilimumab in 834 SCLC patients after four dosages of chemotherapy was examined in the phase III CHECKMATE 451 trial [128]. The test arm was treated with nivolumab every two weeks and four sessions of nivolumab with ipilimumab every three weeks, and the results were compared with the placebo arm. Both ipilimumab and nivolumab (9.2 vs. 9.6 months) and nivolumab alone (10.4 vs. 9.6 months) did not significantly differ from the placebo in OS. Relative to placebo, immunotherapy was associated with a reduced rate of recurrence in both treatment groups (1.7 vs. 1.4 months, 1.9 vs. 1.4 months, respectively). In patients with TMB of 13 mutations per mega base, nivolumab with ipilimumab showed a trend toward OS advantage. The rate of adverse events in the consolidation arm of nivolumab plus ipilimumab, in the nivolumab arm, and the placebo arm was 52.2%, 11.5%, and 8.4%, respectively.

CTLA-4 inhibitor (ipilimumab) efficiency in ES-SCLC in combination with first-line therapy was studied in ICE and CA184-041 phase II studies and CA184-156 phase 3 study. ICE study consisted of 42 patients in the single arm who received ipilimumab 10 mg/kg together with etoposide and carboplatin. PFS was the main objective of the study, but it was not successfully achieved (median PFS 6.9 months). The chemotherapy drug paclitaxel plus carboplatin was used in the CA184-041 phase II study, which involved 130 patients, and they were randomly grouped into three: the first group received ipilimumab (10 mg/kg) together with chemotherapy; the second group (the staged arm) received ipilimumab along with chemotherapy; the third group received chemotherapy plus placebo. Only a PFS difference (6.4 months vs. 5.3 months) and a statistically insignificant OS difference (12.9 months vs. 9.9 months) were seen between the second arm of the study and the placebo group. There was no discernible difference between the third arm and the placebo. CheckMate 032 phase I/II comprises patients with disease recurrence after first-line platinum-based therapy. Both randomized and non-randomized arms were included in the study. In the non-randomized arm, four cycles of nivolumab in combination with ipilimumab were administered to 61 patients, and nivolumab was tested on 98 patients every two weeks. Interestingly, the randomized arm contained 95 and 147 patients, respectively. Interestingly, nivolumab alone had RR for the second line or later that were approximately 11–12% in the pooled cohort, whereas a doubled response in the combination arm was noted [129]. 

## 7. Cancer Immunotherapy Approaches

Various immunotherapy approaches include cytokines, ICIs, monoclonal antibodies (mAbs), vaccinations, and adoptive cell transfer (ACT) [130]. Immunotherapy shows a competitive advantage over other LC therapeutic approaches but determining personalized treatment strategy and when and how to target tumor cells remains a challenge [131]. The immunotherapy-based clinical trials with their central design and descriptions are listed in Table 3.

### 7.1. Checkpoint Inhibitors

The range of standard therapeutic options available to patients with metastatic NSCLC has recently been expanded by the use of ICIs that are used alone or in combination with therapeutic strategies including anti-angiogenic antibodies and chemother-apy [133]. ICIs that target PD-L1 or PD-1 and the CTLA-4 to treat advanced NSCLC are approved by FDA [134,135]. Immune checkpoint drugs work primarily by blocking the immune inhibitory signal pathways activated by the interplay of PD-1 and its ligand PD-L1, restoring the normal capacity of T lymphocytes to destroy tumor cells [136]. 

It was verified that for 12 types of human malignancies, ICIs have a significant curative effect (melanoma, colorectal, head and neck, esophagus, bladder, gastric, hematology, breast, ovarian, lung, and pediatric cancers). The likelihood of high neoantigen content and the TMB vary greatly between various cancer types. Studies have addressed that combining PD-L1 and TMB as composite biomarkers possess a better predictive capacity for the combination strategy [137,138]. Tumors with a high TMB are more susceptible to anti-PD-1 therapy because it can increase T cell response efficacy [139]. IgG1 and IgG4 antibodies that can target the PD-L1/PD-1 axis for NSCLC in combinational first line therapy includes atezolizumab, avelumab, durvalumab, cemiplimab, nivolumab, and pembrolizumab [140]. Recently, well-studied immune checkpoint inhibitions including blockade with mAbs, are utilized as a treatment for several illnesses, including malignancies with diverse origins. Combinatorial therapy, which uses mAbs and other medications at a tolerable dose can prolong survival in NSCLC patients [141].

BMS-936559 is the first IgG4 mAb that was directed against PD-L1 and showed potential therapeutic effects in NSCLC. In a phase I study to test BMS-936559 in 49 NSCLC patients, an ORR of 10% was demonstrated, out of which; 12% had stable disease (SD) at 6 months; 31% had PFS at 24 weeks, and an RR that was irrespective of histology [142]. NSCLC patients at stage IV received the PD-1 inhibitor nivolumab in the second line and pembrolizumab in the first line, had up to 16% and 31.9% of patients living for five years, respectively [117,143]. PD-L1 is better used as a biomarker in clinical practice, yet its accuracy in predicting outcomes is not 100% [144]. 

CD28 possesses competitive binding with CTLA-4 on the activated CD8+/CD4+ T cells for their natural interaction with B7 molecules [136]. An anti-CTLA-4 mAb (IgG1), Ipilimumab, inhibits the binding of CTLA-4 to its ligand. Patients who underwent chemotherapy with ipilimumab had a greater OS than those who had chemotherapy alone (12.2 vs. 8.3 months) [145]. Nonetheless, blockade of CTLA-4 pathway causes more toxicity than compared to siege of PD-1/PD-L1 pathway [146].

Phase I Keynote 001 trial, showed that patients with treatment-naive NSCLC and PD-L1 TPS of not more than 50% benefited from pembrolizumab, achieving a 58.3% RR, 12.5 months of median PFS, and a 24-month OS rate of 60.6% [147]. Pembroli-zumab was approved by FDA for patients that exhibit a higher TMB with any tumor histotypes as a response to the KEY-NOTE-158 trial [138]. In phase II trial of 21 advanced NSCLC patients who were refractory or had advanced to at least first-line treatment, cemiplimab was assessed and observed to have an ORR (6/21) of 28.6% and disease control rate (DCR) (12/21) of 57.1% along with grade 3 TEAEs. Within both non-squamous and SQCC, the ICI, nivolumab enhanced the median OS when compared to docetaxel [148]. 

Access to non-invasive cancer-specific mutations is possible through the new liquid biopsy method, which can be used to profile circulating tumor DNA (ctDNA) and RNA (ctRNA) released into the blood by tumor cells [149]. A low-risk category can be identified by decreased ctDNA levels or its degradation with ICI therapy [150]. 34 NSCLC patients receiving ICI treatment were monitored using ctDNA analysis and blood kirsten rat sarcoma virus (*KRAS*) mutation detection. Before treatment, finding a KRAS mutation had no obvious prognostic significance, but later on, significantly shorter PFS and a shorter OS were linked to a novel *KRAS* mutation that emerged after 3 to 4 weeks of therapy [151]. 

### 7.2. Monoclonal Antibodies

Monoclonal antibody therapy has proven to be a successful alternative that produces good results with reduced adverse effects. Four mAbs, pembrolizumab, cetuximab, bevacizumab, and nivolumab are licensed by FDA to treat NSCLC in recent years. These clinical trials showed potential benefits for NSCLC. Patients with malignancies linked to EGFR mutations are given erlotinib afatinib, and gefitinib; while a kinase inhibitor, crizotinib has been authenticated to treat tumors containing ALK changes [152]. Cetuximab targets the EGFR, which is present in 80%–85% of people with NSCLC [153,154,155]. According to earlier research, adding chemotherapy to this treatment increases the likelihood of survival. For early-stage NSCLC, cetuximab has shown hope as neoadjuvant therapy when combined with other medications, like docetaxel and cisplatin [156]. Bevacizumab, a humanized anti-VEGF monoclonal, is the first medicine to receive approval against tumor angiogenesis. In NSCLC, the expression of EGFR /human epidermal growth factor receptor 1 (HER-1) and VEGF are linked to poor prognosis. Advanced NSCLC can be treated more effectively by combining various mAbs that have direct effects on tumor cells. Phase I of Pembrolizumab clinical trials produced positive outcomes and reduced tumor size in 18% of advanced NSCLC patients who no more responded to chemotherapy [157]. In 2015, FDA authorized pembrolizumab for LC patients as second-line therapy [158]. According to combination trials primarily involving NSCLC patients, Figitumumab, an anti-IGF-1 receptor (IGF-1R) mAb was shown to be effective in a phase I and randomized phase II research when combined with paclitaxel and carboplatin [159]. 

### 7.3. CAR-T Cell Therapy

Chimeric Antigen Receptor- T cells (CAR-T) are genetically modified T cells that express synthetic CAR vectors to recognize and attach to antigens (like CD19) on tumor cells [160,161]. The exact costimulatory molecules vary most significantly between CAR generations, and the fifth generation of CARs is now being tested [162,163]. NSCLC is the solid tumor type on which most CAR-T cell research is concentrated. In the EGFR based CAR-T therapy of NSCLC, two patients demonstrated partial response, and five patients displayed stable illness; has demonstrated additional need for CAR-T cells to treat NSCLC in the future. EGFR, prostate stem cell antigen (PSCA), mesothelin (MSLN), carcinoembryonic antigen (CEA), mucin 1 (MUC1), PD-L1, inactive tyrosine-protein kinase transmembrane receptor (ROR1), CD80/CD86, HER2 are the antigens most frequently targeted in NSCLC. More than 60% of NSCLC mutations are related to neovascularization, tumor growth, and metastasis. Recombinant anti-EGFR CAR-T cells exhibits cytolytic activity specifically against tumor cells that are EGFR-positive [164]. 

At Sun Yat-sen University, C-X-C chemokine receptor (CXCR) type 5-modified anti-EGFR CAR-T cells are evaluated in a phase I clinical trial to test their effectiveness and reliability in treating advanced NSCLC patients with EGFR mutations (NCT04153799). It was found that patients can easily administer anti-EGFR CAR-T cell perfusions three to five days at a time. Therefore, although more clinical research is required to support these findings, anti-EGFR CAR-T cells have turned up to be effective in treating NSCLC patients who possess the EGFR mutation [164]. Lower chance of survivability and tumor aggressiveness are linked with high expression of MSLN in individuals with early-stage lung ADC [165]. Benefits of anti-MSLN CAR-T cell treatment for NSCLC were evidenced by the management of mRNA-engineered T cells intravenously that allowed for transient expression of an anti-MSLN CAR, but metastatic tumors in NSCLC were not revealed [164]. A transmembrane glycoprotein called MUC1 is overexpressed in numerous cancer forms, including NSCLC. The effectiveness and security of anti-MUC1 CAR-T cell therapy combined with PD-1 deletion are being evaluated in Phase I/II clinical trial in advanced NSCLC patients (NCT03525782) [164]. In patient-derived xenograft (PDX) model, anti-MUC1 CAR-T cells were unable to effectively slow down the development of an NSCLC tumor mass [166]. Anti-MUC1 and anti-PSCA CAR-T cells can potentially work together to treat NSCLC more efficaciously. A phase I research (NCT03330834) to assess the safety, tolerance, and engraftment potential of autologous CAR-T cells that target CD80/CD86 and PD-L1 is used to treat recurrent or refractory NSCLC. In a phase I study of patients with PD-L1-positive advanced NSCLC, anti-PD-L1 CAR-T-cell treatments are tested for their dependability and effectivities [164]. Anti-ROR1 CAR-T cells used organoid tumor models and successfully killed NSCLC and triple-negative breast cancer (TNBC) cells. Moreover, tyrosine kinase-like ROR1 is an orphan receptor found in both NSCLC and TNBC. On that account, anti-ROR1 CAR-T-cell therapy emerges as an advanced strategy for the treatment of NSCLC [167].

The toxicity of CAR-T cells in clinical settings is still a concern, although they offer a potential method for treating NSCLC. As many TAAs are not tumor-specific, CAR-T cell nonspecific interaction with normal cells may lead to toxicity. A longer extent of in vivo efficacy, stronger ability to bind to targets, and a faster curative effect on NSCLC are the advantages of CAR-T over conventional therapy. The safety and effectiveness of conventional CARs have also been enhanced by the development of multi-target, drug-inducible, dual-target switchable, universal, and inhibitory CARs [168,169,170,171].

### 7.4. Lung Cancer Vaccines

Patients receiving chemotherapy for advanced-stage disease frequently possess drug-induced harsh side effects and have short-lived responses, especially when combination chemotherapy is given for an extended period. Vaccines may offer a therapeutic potential when incorporated into treatment plans soon after the first round of chemotherapy. A few phase III trials are being conducted to evaluate these vaccinations [172]. After definitive treatment, patients with minimally recurrent disease are likely to benefit more from vaccinations and may have long-lasting therapeutic effects (Figure 2).

#### 7.4.1. Belagenpumatucel-L Vaccine (Lucanix)

One of the negative diagnostic factors of NSCLC, TGF-β, is identified to present in elevated levels in various malignan-cies including LC [173]. Preclinical research has demonstrated that the immunogenicity of tumor vaccines is raised by TGF-β2 suppression which serves as a repository for many TAAs [174]. An allogeneic tumor cell, gene-modified vaccine called Lu-canixTM (NovaRx, San Diego, CA, USA) is tested for four distinct NSCLC cancer cell lines that suppress TGF-β2 expression and boost immunogenicity by producing a TGF-β2 antisense gene [175,176]. At the injection site, the cause of immune suppression for vaccine is expected to be due to reduced TGF-β2 expression in the vaccine. The idea is that injecting downregulated TGF-β2 into allogeneic tumor cells will improve local immune identification and activate effector cells, triggering a systemic immune response that can target the patient’s original tumor [177]. Lucanix to placebo, a phase III trial as maintenance therapy did not find any differences in OS or PFS, documenting 96 NSCLC [178,179].

#### 7.4.2. MAGE-A3

MAGE-A family consists of more than 60 genes and most of them are positioned on X chromosome to which belongs the melanoma antigen A3 (MAGE-A3) [180]. TAA which are expressed explicitly on tumor cells include MAGE protein. It is expressed in male germ cell lines, but not in other normal cells, and due to the lack of MHC they are incapable to hand-out MAGE-A antigens to the immune system [181]. The MAGE-A3 expression is considered directly proportional to the cancer progression. As the disease spread, MAGE-A3, whose normal expression level was 35% in NSCLC changed from 30% in stage I patients to 50% in stage II patients indicating their correlation with poor prognosis [182,183]. Cancer germline genes are important factors in determining the immortality, tumorigenesis, invasiveness, metastatic, and immune evasion capacity of tumor cells [184]. The tumor cell morphology, adhesion and migration can be altered by downregulating expression of cancer-germline gene [185,186]. Comparison of postoperative injections of MAGE-A3 recombinant protein coupled with an adjuvant system in 182 patients of a randomized phase II trial resulted in an outcome that was statistically insignificant, although the survival benefits of MAGE-A3 were sufficiently robust to initiate a Phase III assessment [181,187]. To establish a predictive signature that matches up with the treatment of MAGE-A3 antigen-specific immunotherapy, the gene expression profiling of tumors before treatment was analyzed [188]. In a population with a predictive genetic signature, the risk of recurrence was reduced by 43% after treatment with MAGE-A3. Another phase III LC trial commenced in 2007 with NSCLC patients recruited from 33 countries. In patients with MAGE-A3–positive stages IB, II, and IIIA, the efficacy of MAGE-A3 antigen-specific cancer immunotherapeutic (ASCI) agents were tested [189]. A total of 13,849 patients were screened, of which, 4210 had a MAGE-A3 positive tumor and 2272 patients were selected by randomization and then treated. A gene signature that aided in clinical benefit to MAGE-A3 was not observed because disease free survival (DFS) did not escalate on NSCLC in either the overall popula-tion or in NSCLC patients who did not receive ACT after treatment.

#### 7.4.3. CIMAvax-EGF

CIMAvax-EGF is EGF based recombinant vaccine conjugated chemically onto p64K and emulsified in Montanide ISA 51 as adjuvant [190,191]. Antibodies against EGF are produced as part of mechanism of action of CIMAvax EGF, blocking EGF-EGFR interaction, and inhibit EGFR phosphorylation that results in EGF withdrawal [192]. A total of five phase I/II, a randomized controlled phase II and another randomized controlled phase III clinical trials have been per-formed since 1995. The first five trials were modeled to optimize the formulation, dosage and timing of the vaccine concerning immunogenicity and safety [193,194]. The pioneer step that attested the immunogenicity and the feasibility of inducing an antibody titer against autologous EGF was conducted in patients with colon, lung, prostate cancer or stomach [190,195,196]. The phase II randomized trial could not show that the vaccinated cohort had an advantage in OS [197]. In patients younger than 60 years of age and with GAR compared to PAR in the vaccinated group, a significant trend toward a survival advantage was seen. However, an advantage in OS for vaccinated patients in the phase III trial was not observed. As a possible diagnostic and predictive factor, baseline serum EGF levels were revealed [198]. The link between CIMAvax-EGF and chemo-combination therapy was disclosed by the Pilot 5 study, regardless of the small sample size. It revealed the potential rise in vaccine immunogenicity when conjugated to a cytotoxic systemic treatment [197]. 

#### 7.4.4. Racotumomab

Racotumomab, the monoclonal anti-idiotypic murine IgG1 vaccine [199], causes the N-glycolyl GM3 (NeuGcGM3) ganglioside, almost undetectable in normal cells but seen in some tumor cells to provoke a particular humoral and cellular immune response. Their overexpression remains a target for cancer therapy, and is linked to altered cell proliferation, tumor spread, angiogenesis, and immunological tolerance [200,201]. According to a meta-analysis of stage III and IV NSCLC comprising 26 studies and 7839 patients, racotumomab and pemetrexed maintenance therapy were found to be successful in terms of DFS and OS [202]. A phase II/III trial with advanced NSCLC in 87 patients led to the conclusion that the product is effective, safe, and showed an increase in OS and PFS. The trial reached a certain conclusion from the results: drug induced only minor adverse effects like localized injection site responses, bone pain, coughing, and asthenia [203]. In 71 NSCLC patients for whom surgical treatment was not a choice, racotumomab was used as a second line therapy, and has shown to enhance OS in stage IV NSCLC patients [204].

#### 7.4.5. TG4010

MUC1 is a TAA expressed by many solid tumors, including NSCLC. TG4010 targets MUC1 TAA and IL-2 [205]. The MUC1 protein is upregulated in LC and appears abnormally glycosylated, making it an immune target as glycosylation is the reservoir for new antigens. In a randomized, multicenter, phase II study with advanced NSCLC patients, TG4010 was administered in combination with platinum-based chemotherapy (cisplatin/vinorelbine) wherein the ORR and time to progression were 29.5% and 4.8 month in arm 1, respectively. In 2011, the study was further extended to open-label randomized trial III, which comprised 100 and 48 patients expressing MUC1 in their tumor with stage IIIB and stage IV NSCLC, respectively [206]. In this study, to predict the response to TG4010, frequency of CD56+, CD16+, and CD69+ cells were considered as biomarkers as they correspond to the phenotype of activated NK cells in PBMCs. 

#### 7.4.6. Vaccine Delivery Vehicles

Even though immunotherapy is a successful treatment option for NSCLC, there are still several limitations such as tumor penetration, low efficacy, high toxicity, issue with optimization of synergistic treatment, and off-target effects re-main. Effective delivery methods can overcome all of these restrictions [207,208]. A crucial technique for creating a com-prehensive therapeutic approach is making of an effective delivery system. For instance, therapeutic nanoparticles (NPs), are used to jointly administer chemo-immunotherapy regimens such as IL-12 and doxorubicin to achieve effective delivery into the tumor [209]. 

##### Nanoparticle-Based Delivery

Small molecules and their antagonists, for example, CpG oligodeoxynucleotides, inhibitors of TGF-β, IL-2, anti-PD-1 mAbs, antibodies and their fragments, peptides and proteins, can be delivered by NPs [210]. A few options available for NPs are nucleic acid nanotechnology, liposomes, dendrimers, inorganic nanocarriers polymeric systems, and exosomes. Delivery of cancer immunotherapies by NPs improves drug penetration, antitumor efficacy, synergetic effect of treatments and drug retention [211,212]. Additionally, by employing techniques like drug efflux pump modulation and administering numerous medications, NPs can overcome chemotherapeutic resistance.

##### NP-Loaded Small Molecules

Advanced LC tumors require a standard of care of combination chemotherapy and radiotherapy and additional cut-ting-edge treatments like immunotherapy or tailored medication. The most recent advancements in nanomedicine for the treatment of LC are magnetic NPs (such as: GastromarkVR, LumirenVR, Feridex, EndoremVR, and I.V.VR), liposomes and solid lipid vaccines (metal NPs (zinc oxide NPs, titanium dioxide NPs, gold NPs, iron NPs, silver NPs, and cerium NPs), cisplatin-loaded lipid-chitosan hybrid NPs, taxane class of NPs, virus NPs (TMV-cisplatin VNP complex, DOX-PhMV-PEG, Trop2CD40L VLPs) and polymer NPs (doxorubicin-conjugated InP/ZnS QDs) [213]. 

##### Extracellular Vesicles

Tiny, membrane-enclosed vesicles released by living cells that can carry a variety of lipids, nucleic acid species, and proteins are called extracellular vesicles (EVs) [214]. Anti-tumor immune responses are regulated by EVs, which assist in cell-cell interaction within TME [215]. Recently EVs have been identified as vehicles for various bioactive agents of cancer immunotherapy. For instance, designed EVs developed from fibroblast-like mesenchymal cells carrying short hairpin RNAs (shRNAs) or small interfering RNA (siRNAs), that target KRAS, are found to improve anti-pancreatic cancer activity and also raise mouse OS rates. The advantages of using EVs as a delivery platform are numerous and include their capacity to overcome environmental obstacles, inherent cell-targeting abilities, and circulatory stability [216].

##### Antigen-Mediated Delivery

CHP-NY-ESO-1 vaccine system aids in enhanced immune response by providing targeted delivery and is one of the important antigen delivery approaches in cancer immunotherapy. The polysaccharide nanogels of cholesteryl group-modified pullulan (CHP), that regulates TAMs are used to deliver the cancer-testis antigen, New York esophageal squamous cell carcinoma (NY-ESO)-1. The survival rate in patients with NSCLC found to be dose dependent on CHP-NY-ESO-1 in which a higher dose produces a longer survival rate. Targeted delivery and an improved immune response are possible with this approach [217].

##### Cell-Based Delivery

A common procedure of cell-based therapy involves ACT, CAR-T, and TIL therapy [218]. CAR-T cell treatment is a highly successful immunotherapy in the fight against blood malignancies and cancers that are refractory. The FDA has ap-proved a number of CAR-T cell-mediated immunotherapy products. Breyanzi can be given to patients with refractory or relapsed large B-cell lymphoma, Abecma and Carvykti for refractory or relapsed multiple myeloma patients, Yescarta for large B-cell lymphoma patients, and Tecartus for patients with refractory or relapsed mantle cell lymphoma [219]. In addition, drugs can act as nanoscale medicines that do not possess carriers advised for cancer care. Types of nanomedi-cines free of cargo are drug nanocrystals, antibody-drug conjugates (ADCs), drug-drug conjugate NPs and prodrug self-assembled NPs. For instance, to boost drug lodgment to tumors and kill both non-cancer stem cells (non-CSC) and CSCs, the medication SN38 (7-ethyl-10-hydroxycamptothecin) has been combined with the pH-responsive prodrug (PEG-CH=N-Doxorubicin (DOX) [213].

## 8. Combinatorial Approaches to Lung Cancer-Immunotherapy

The traditional therapeutic methods continue to be the first choice of cancer treatment; however, the success of these conventional regimens is limited by their unpleasant side effects, late illness detection, and deadly reversion as a resistant micro-metastatic disease. Hence, the gold standard of care for patients whose disease is not caused by a genomic change in NSCLC is PD-1 and PD-L1 inhibitors alone or given along with chemotherapy, radiation, or other ICIs [220]. 

### 8.1. a. Targeted therapy

Targeted therapy is the best option for those with irreversible LC and a driver gene mutation [221]. FDA-approved first-line medicines for metastatic NSCLC patients whose tumors possess *EGFR* exon 21 L858R mutations or *EGFR* exon 19 deletions are gefitinib, erlotinib, afatinib, dacomitinib, and osimertinib. *EGFR/HER2* exon 20 mutations containing patients responded well to the two newly developed targeted medications such as TAK-788 and poziotinib [148]. A novel fusion oncogene echinoderm microtubule-associated protein-like (EML4)-ALK is seen in young and non-smokers, and 2–7% of patients with advanced NSCLC [221,222]. AMG510 is a tiny, effective drug that locks *KRAS* G12C in an inactive, GDP-bound state. Another *KRAS* G12C inhibitor, MRTX849, has shown promise in treating advanced solid tumors with *KRAS* G12C mutations [148]. IFN, which activates a variety of immune cell types, is found to be produced more abundantly by combining systemic IL-12 with trastuzumab and paclitaxel in a phase I clinical trial, and this further enhanced NK cell activation [223]. Their favorable correlation with clinical response potentiates the use of combinatorial strategies.

### 8.2. b. Chemotherapy 

Platinum-based chemotherapy, with roughly one-year median survival, is crucial in treatment methods for patients without a driver gene mutation [224]. In multiple randomized controlled trials in NSCLC and SCLC, chemotherapy plus ICI showed better results than chemotherapy alone [144]. The CheckMate-9LA study showed that chemotherapy for a shorter period, along with inhibitors of CTLA-4 and PD-1 is feasible without sacrificing efficacy [225,226]. Ipilimumab, a CTLA-4 inhibitor, was given to NSCLC patients included in the phase II study either in a steady regimen in cycles 3 to cycle 6 of chemotherapy or concurrently in cycles 1 to 4 of six chemotherapy cycles, followed by ipilimumab maintenance [227]. In the KEYNOTE-021 study, the carboplatin-pemetrexed-pembrolizumab combination had an ORR of almost 70%, testing the activity of pembrolizumab with carboplatin and paclitaxel, vs. carboplatin, paclitaxel, and bevacizumab, or carboplatin and pemetrexed [228]. In a recent phase Ib study, the ICI durvalumab (PD-L1) and tremelimumab (CTLA-4) were coupled with the chemotherapy drug cisplatin-pemetrexed. Two ICIs combined with conventional chemotherapy appeared to be well tolerated, which revealed an ORR of almost 50% without specific hazard effects [229]. Quadruple combination therapy using bevacizumab, platinum-based chemotherapy, and atezolizumab was effective in treating patients with NSCLC who were oncogene-dependent [142,230].

### 8.3. c. Radiotherapy

Immunomodulatory effects of radiation include tumor-associated dendritic cell activation, improved T-cell action on tumors, altered TAM polarization, reduced immunosuppressive stromal cells, and induced immunogenic cell death [231]. The possibility of toxicity, particularly pneumonitis, when radiation and ICI are used together is a significant issue [232]. It has also been demonstrated that high-dose ionizing radiation increases PD-L1 expression, and inhibition of PD-L1 improves the effectiveness of radiation by a CTL-dependent mechanism [233]. Anti-CTLA-4 and anti-PD-1 therapy with combined radiation enhance the number of tumor-specific T cells, and it is noted that radiation-induced immune responses can have anticancer effects [145]. Radiation, when combined with CTLA-4 inhibition, produced diversification of the T cell receptor repertoire of TILs and molded the repertoire of T cell clones that are expanded, which is consistent with an effective vaccination [234]. It was discovered that overexpression of PD-L1, which causes T-cell exhaustion, is a mediator of radiation resistance and CTLA-4 blocking [235]. 

### 8.4. Emerging Combination Strategies in Adoptive Cell Therapy

Reactive T cells are extracted from patients, amplified *ex vivo*, and returned to the patient, which will aid host immunity for battling the disease [236] and is a more direct, less expensive, and effective technique to produce potent immunotherapeutic molecules [237,238]. Adoptive immunotherapy comes in a variety of forms that are employed in clinical trials, including lymphokine-activated killer (LAK), cytokine-induced killer (CIK), γ δ T-cells, TAA, TIL, and specific CTL [239]. NK-resistant tumor cells can be killed by LAK cells produced by lymphocytes exposed to IL-2. For patients with metastatic melanoma, TIL-based adoptive treatment has shown promising anticancer activity in a great number of clinical trials [240]. LAK possesses a powerful ability to lyse tumor cells that have survived after receiving cisplatin treatment and impart cytotoxic effects on A549 cells [241]. IL-2 and LAK cells coupled with chemotherapy or radiotherapy enhanced the survival of patients following surgical resection of primary LC, with 5- 9 years of survival rates in the immunotherapy group. Immunochemotherapy and immunoradiotherapy have shown OS of 54.4% and 52.0% in patient groups and 33.4% and 24.2% in the control group.

PD-L1 expression and the degree of CD8+ T cell infiltration were affected by concurrent chemoradiotherapy, and both affected the prognosis of NSCLC patients [242]. 22.4 months and 14.1 months was the median survival in the immunotherapy group and in the control group in a randomized trial, respectively, that added IL-2 and TIL to standard chemotherapy/radiotherapy after surgery. For stage III cancer patients, local relapse was significantly decreased in the immunotherapy group but not in the distant relapse [243]. After concurrent chemoradiotherapy (CCRT) treatment, Yoneda et al. noted that CD8+ TIL with increased density is a good prognostic indicator for locally progressed NSCLC [244]. The heterogeneous T-lymphocytes, known as CIK cells, have unlimited MHC-wide anticancer activity and a mixed NKT phenotype [245]. Seven ongoing CIK trials are being conducted, and a recent study among early-stage patients revealed that the 2-year OS rate could be increased when chemotherapy is coupled with DC-activated CIK cells, but the DFS rate does not get affected [246]. In patients receiving CIK in addition to chemotherapy, Wu et al. noticed that PFS (*p* = 0.042) and OS (*p* = 0.029) were longer [247]. Anti-γ δ TCR antibodies can be used to grow γ δ T-cells in vitro; this method may be more effective than employing phosphoantigen-expanded γ δ T-cells because the expanded γ δ T-cells have a longer in vivo survival time [248]. The effectiveness of CAR-NK cells to precisely identify tumor antigens has been demonstrated in numerous preclinical and clinical investigations, and therefore, adaptive NK-cell-based therapies can benefit from the features of recombinant CARs [249]. The phase-I trials, which were performed on patients with advanced and recurrent NSCLC revealed that activated NK cells are safe, well tolerated, and exhibit no substantial toxicity. Patients with LC were found to benefit clinically from repeated infusions of in vitro-activated HLA-mismatched NK cells when combined with conventional treatment [250]. In late-stage LC patients receiving vinorelbine-platinum chemotherapy, administration of a combination of immune cells, peptide-pulsed DCs and CIK were found to lessen the side effects of chemotherapy and increase patient survival [251].

Due to the viability and functionality of transplanted cells, which are instantly rejected after injection, adoptive treatment applications are restricted [252]. Nanocarrier systems based on innovative technologies can overcome these constraints. With tailored NPs, better adoptive T-cell treatment against prostate cancer was described [253,254]. Maleimide was surface conjugated to liposomes before being evaluated in a metastatic melanoma murine model. It was discovered that animals treated with NP-conjugated T-cells completely cleared the tumors, but non-conjugated T-cells slightly improved the survival of untreated mice [253]. These studies report that we can improve the effectiveness of ACT by delivering T cells using biomaterials [252]. Hence, according to recent clinical research, adoptive immunotherapy is effective for treating NSCLC and aid in an improved overall outcome and minimal toxicity.

## 9. Lung Cancer Preclinical Models in Testing Immunotherapy 

### 9.1. 3D Models 

TME contain many factors that can affect the evolution and development of tumor. It can affect the metabolism, vascularization, and immune system of tumor tissues. It is comprehensive to study tumors using 2D culture models due to their inability to mimic the whole TME. 3D models can exhibit the proliferation, activation, and immuno-modulatory abilities of tumor tissues more than 2D culture models [255] (Table 4).

#### 9.1.1. Cell-Lines

For the past 40 to 50 years, the spheroid model has been regarded as the gold standard among 3D in vitro models. Spheroids are 3D-based collections of cells along with ECM. To a certain extent, they can mimic the structure and metabolism of the tissue from where they originated. Based on cell genesis and culture methodologies, several spheroids may be defined, including multicellular tumor spheroids (MTCS), tumorospheres, tissue-derived tumor spheres (TDTSs), and organotypic multicellular spheroid/organoid (OMS) [256]. The most popular and thoroughly defined model, MTCS, is frequently produced from primary cell or cell line suspension. The MTCS model accurately depicts the diffusion and exchange of oxygen, nutrients, and other soluble factors. Due to its remarkable repeatability and relatively low cost, it is now the most used model for evaluating immunotherapeutic techniques [257].

Recently, inhibitors for immune checkpoints have been used for therapy in the fourth stage of NSCLC. In several studies, patient-derived NSCLC spheroids were used as a model to predict immune checkpoint treatment sensitivity. Spheroid heterotypic models are frequently employed to simulate the immunological microenvironment. The anticancer effect of CTLA4, PD L1, and PD1 blockage are mediated by immune cells in spheroids; also, they are characterized by increased CD8+ T cells and pro-inflammatory cytokine expression. There was no correlation analysis performed with patients’ outcomes in this model. Only a single clinical study on PD 1 therapeutic effect shows consistency in this model. Therefore, the predictive values are unclear, and this model cannot be considered a valid model in immunotherapy. New studies in this area are necessary to evaluate patient-derived spheroids’ prognostic role in testing immunotherapy [258].

#### 9.1.2. Patient-Derived Lung Cancer Organoid

Scientific attention has switched to 3D cell culture techniques since 2D cell cultures only preserve a minimal similarity to their parent tissue. Patient-derived cell populations may be expanded in a 3D ECM to form organoids, distinguished by their ability to form structures that resemble the tissue from which they were obtained. Numerous tissues, most notably the lung, have been used to create tumor organoids [259]. LCOs require a specific concoction of growth factors and inhibitors, as well as a 3D matrix for support, most frequently Matrigel. Although the formulation of this growing medium differs between laboratories, all formulations include components that support the preservation of lung stem cells. Growth factors included are either EGF or members of the fibroblast growth factor (FGF) family, as well as inhibitors/activators of specific pathways, namely TGF-β and Rho-associated protein kinase and Wnt. The B27 supplement used as a substituted serum in neuronal cell cultures is the only factor common to all reported LCO-specific media formulations. LCOs may be created from tumor biopsies or resections that show a high rate of success in both short- and long-term cultures [260,261]. Li et al. developed a patient-derived tumoroid assay and the whole sequence to screen afatinib’s effect in EGFR-mutated patients [262]. Using transcriptomics, Peng et al. compared the primary tissues of SQCC or ADC to the equivalent tumoroids. Comparing tumoroid models to cancer cell lines or PDXs, tumoroid models showed a greater transcriptional accuracy [263]. Another study conducted by Ma et al. in these same tumoroids identified the genes (*CDK1, CCNB2*, and *CDC25A*) implicated in the NSCLC tumor formation [264] (Table 5).

#### 9.1.3. Patient-Derived 3D Models 

PDOs enable the 3D cultivation of malignant cells from the primary tissue, resulting in stromal destruction and immunological compartments. Tumorospheres are clonal representations of spheroids or organoids. However, this spheroid model is only suitable for CSC research since it cannot capture the diversity of cell types found in the TME. TDTSs are made from the patient tumor tissue that enzymes have digested. This model can preserve only the interactions between tumor cells, not between the stroma and tumor cells. TDTSs produce small replicas of vascularized tumor regions. OMS preserve the cellular heterogeneity and origin tissue architecture, making them the most accurate replicas of the parental tumor. OMS is useful for biomarker and medication testing since they are reliable preclinical patient models. PDOs replicate the gene and protein expression of the real biopsies. Several models are available, including those developed from NSCLC, clear cell renal cell carcinoma (CCRC), melanoma, and glioblastoma [255].

**Table 4 vaccines-10-01963-t004:** The main studies conducted in the field of lung cancer using spheroids and organoids models.

Models	Source	Application	Reference
spheroids	Resection	Lung cancer stem cells’ identification and description; production of Xenografts that replicate the parental tumor’s histology	[265]
	Resection, pleural effusion	Technique to expand cancer cells from the lungs of patients	[266]
	Core needle biopsy, surgical biopsy, pleural effusion	Drug testing	[267]
Organoids	biopsy	Drug testing and evaluation of immune cell populations penetrating cultured tissues	[268]
	Resection/PDX	LCOs’ long-term growth, confirmation, and drug screening	[269,270]
	biopsy	For use in immuno-oncology research and testing for customized immunotherapy, a new approach for maintaining endogenous tumor-infiltrating cells has been developed.	[271]
	Resection/biopsy	Biobanking, drug testing	[272,273,274]
	Resection/biopsy	Examining and blocking regulators of mitochondrial fission in various tumor organoids	[275]
	Pleural effusion	drug testing and the establishment of an LCO culture system from pleural effusions	[276]
	Resection/biopsy	Analyzing several techniques to determine the tumor purity of organoids created from intrapulmonary tumors	[277]
	PDX derived from biopsies	Drug testing and organoid creation using PDXs from SCLC biopsies	[278]
	Resection/biopsy	Analyses of pathway inhibitors found by single-cell proteomics	[279]
	pleural effusion	Targeted drug testing and LCO production and characterization	[280]

**Table 5 vaccines-10-01963-t005:** Currently used lung organoids and tumoroids models in NSCLC research.

Tumor Type	Model	Application	Reference
NSCLC (Non-small cell lung carcinoma)	Lung cancer organoids	Drug screening	[269]
	NSCLC organoids	Drug screening	[270]
	Patient-derived organoids model	Genomics, production of treatment outcomes	[274]
	Lung ADK (LADC)-derived organoid model	Drug screening, biomarker development, and living biobank	[273,274]
	Patient-derived tumoroids (PDTs)	PDTs are developed to be used in microfluidic devices for drug screening and mimic the cancer vascular network.	[281]
	Patient-derived lung cancer organoids	Patient-specific medication screening and support for the xenograft model from a living biobank	[272]
	Patient-derived tumoroids (PDTs)	creating new cell lines	[282]
	Patient lung-derived tumoroids (PLDTs)	Drug screening	[283]
	Lung cancer organoids	Personalized medicine	[277]

#### 9.1.4. Organ-On-Chip

Although there have been in vitro models of LC for many years, most of them have consisted of monolayer cultures of lung epithelial cells or planar 3D tissue models with air-liquid interfaces. The model can be used to trace pharmacological treatments and immune cells [284]. NSCLC-derived cell lines can be employed in microfluidic models of LC, where the A549 cell line is most often used. A549 cells often acquire the spheroid shape when cultivated in microfluidic devices because they cannot establish intercellular connections. These platforms have been utilized to research a variety of significant issues [285]. The effectiveness of photodynamic treatment was analyzed using A549 spheroids cultivated in microfluidic devices. The ability to precisely manipulate and characterize cell-cell communication in the LC TME has been made possible using microfluidic devices, for instance, to segregate different cell types seen in LC such as CAFs and ECs. NSCLC cell lines were cultured in microfluidic models in various studies to investigate the effects of tumor growth, medication response, and mechanical forces found in the lung, including blood and interstitial fluid flow. The interaction of a lung tumor and bacteria has also been studied using NSCLC cells cultured in microfluidic devices [284]. Using autologous TILs in two studies from the Borenstein group, small primary tumor organoids from NSCLC patient tumor biopsies were cultured. This allowed for the characterization of tumor-immune interactions and the prediction of patient-specific responses to ICI therapies [286,287]. Future developments in microfluidic models of LC should carefully consider the tumor cell type and their source; the inclusion of different cell types such as immune cells, epithelial cells, and/or CAFs; and the biomarkers to represent ECM and hypoxia. Furthermore, the impact of mechanical strain, ECM composition, immune cell phenotype and infiltration, and response to treatment regarding this model is still poorly understood [284].

#### 9.1.5. 3D Bioprinting 

3D bioprinting is a biomanufacturing technology that enables loading live cells, signaling molecules, and biomaterials to create tissue-engineered constructions with precisely regulated tissue architecture. The use of bioprinting technology can produce gradated macroscale designs that imitate the ECM, also boosting both the adhesion and proliferation of various cell types. Constructs created via 3D bioprinting can successfully imitate the TME. Additionally, the capacity to incorporate perusable vascular networks, spatial control of matrix characteristics, and automatic and high-level testing capabilities for identifying metabolic toxins and metabolic parameters are crucial components of bioprinted models. According to their deposition process, droplet, extrusion, and laser-based bioprinting techniques can be categorized based on instrumentation methodologies [288]. A substantial number of biopsy samples from patients with lung cancer were utilized by Mazzocchi et al. to create 3D tumor models containing pleural effusion aspirates and lung cancer spheroids implanted within hydrogel scaffolds [276]. Another study showed that sodium alginate-gelatin hydrogel provides better printability and viability to the cells of NSCLC from patient-derived xenografts and associated fibroblast coculture [289]. The printed tumor models can also be manufactured using spheroids that exhibit tumor-specific markers and are used for drug screening. Wang et al. used A549/95-D lung cancer cells to create 3D bioprinted scaffolds to study metastasis [290]. The bioprinted tumor models developed using patient-derived cancer cells have in vivo-like anatomy [288].

### 9.2. In Vivo Animal Models 

The mouse genome is quite like the human genome and possesses the ease of gene editing, cheap cost, and straightforward feeding. It can imitate several biological properties, including cancer development and metastasis *in vivo.* Four approaches are often used to create animal cancer models, including chemical induction, PDX, genetically engineered mouse models (GEMM), and cell line-derived xenograft (CDX). The chemical induction model is a model induced by chemical carcinogens that can mimic the onset of cancer from the beginning of its carcinogenic process. However, the major drawback is that it takes about 30 to 50 weeks for a tumor to grow. The CDX model, also known as the xenotransplantation model, is created by delivering cancer cell lines subcutaneously to immunocompromised mice. The key merit and demerit of this model include convenience to handle, but the prolonged in vitro culture alters the behavior of original tumor cells. The PDX model was developed as an animal model by introducing patient tumor tissues directly into mice. This model faithfully replicates the histology and genetics of the original tumor [291]. These current models, however, are unable to precisely anticipate how the tumor and immune system of humans would interact among different species of mammals. The humanlike mouse immune system model rebuilds the human immune system by grafting lymphocytes, hematopoietic cells, or organs of humans into mice with a compromised immune system [292] (Figure 3b). By implanting human tumor cells or tissues, it is feasible to understand tumor growth, the human immune system setting, and assess anti-tumor therapy, specifically the impact of immunotherapy and associated processes. The humanized mouse immune system models are categorized into three groups based on how the human immune system was recreated, and it includes human hematopoietic stem cells (Hu-HSCs), human peripheral blood lymphocyte (Hu-PBL), and human bone marrow, liver, and thymus (Hu-BLT) models [291].

#### 9.2.1. Zebrafish Model

Zebrafish is the most popular vertebrate model due to their genome similarity with humans [293]. Transgenic and immunodeficient zebrafish can grow very fast, remain transparent in the adult stage, and have simple gene operations. Translucent embryos can detect and trail cancer cell propagation, metastasis, and spread [291]. Shen et al. developed the LINC00152 knockout xenotransplantation model and demonstrated that LINC00152 silencing might lessen LC cell proliferation and spread [294]. The effectiveness and safety of anti-LC medications may also be assessed using the zebrafish model to determine a more effective course of therapy. Huang et al. employed the zebrafish LC models to assess the impact of DFIQ (quinoline derivative anti-cancer agent) on NSCLC in vivo. Monitoring cell migration, apoptosis, and proliferation revealed that DFIQ might suppress cancer cells. Zebrafish models can accurately reproduce the interactions between TME though they lack lungs. The efficacy and safety of three anti-angiogenic medications have been investigated using a human NSCLC xenograft model. All molecules examined had anti-angiogenic properties and reduced the development of tumors in zebrafish [295].

#### 9.2.2. Patient-Derived Tumor Xenograft (PDX) Model

The Hu-PDX model has been used in several tumor research studies. Lin et al. demonstrated PD-L1/PD-1 targeted immunotherapy in immunodeficient mice using patients’ peripheral blood cells [296]. Rosato and his colleagues constructed a PDX model derived from TNBC patients to test anti-PD-1 immunotherapy as a TNBC preclinical trial [291]. Sanmamed et al. administered immunodeficient mice with gastric cancer patient lymphocytes, then administered mice with transplanted gastric cancer tissue, and then with nivolumab (inhibitor of PD-1) and urelumab (anti-CD137 agonist). This study demonstrated that these drugs could resist tumor growth through the induction of T cells [297]. The present Hu-PDX model does have significant flaws, such as a low modeling success rate, limited lifespan for the humanized immune system, and poor immunological response. The advancement of humanized mouse modeling technologies and the effectiveness and longevity of the immune system should be the main areas of future study [291].

#### 9.2.3. Patient-Derived Orthotropic Xenograft (PDOX) Model

Comparable to the primary site of cancer, an in vivo environment that is favorable for the growth of the tumor can be created through orthotopic transfer of tumor tissue into animal organs. As a result, the PDX model serves as the foundation for the PDOX model. This model is more objective and accurate than the conventional PDX model in simulating the development of human cancers in vivo [298]. Using cervical cancer tissue, Hiroshima et al. created ten cases of the PDX model and eight cases of PDOX models by subcutaneous injection of cervical cancer tissues. The result demonstrated that in 50% of the PDOX models, tumor metastasis was shown but not in the PDX model. According to the data, the PDOX model is more likely to have characteristics of malignant tumors, such as invasion and metastasis, than the PDX model [299]. Hiroshima et al. conducted a study by treating these two models with entinostat medication and found that tumor growth inhibition took place only in PDOX models [291]. In the PDOX model, growth was found in vivo, making it challenging to monitor their progress using conventional detection techniques and even more challenging to identify their metastatic paths [300]. 

#### 9.2.4. Mini Patient-Derived Xenograft (Mini-PDX) Model

This model was created by infusing the patient’s tumor tissue’s digested cell suspension into a microcapsule and implanting the capsule into the animal. The major advantages of this model in the area of drug screening include consistent outcomes with the PDX model, economic, and short time to approach the results [291]. Zhang et al. created Mini-PDX models of pancreatic, gastric, and lung cancers, and PDX model was used as a reference model to assess drug sensitivity. The results indicate that both the Mini-PDX model and the conventional PDX model produce results that are 92% consistent; however, the Mini-PDX methodology requires substantially less time than the PDX model does [301]. This demonstrates that the Mini-PDX model might serve as a useful alternative to the PDX model for assessing cancer treatment. Due to these benefits, the Mini-PDX model can be anticipated as a device that aids in the individualized treatment of cancer patients [291].

## 10. Conclusions and Future Perspectives 

LC continues to pose a serious threat to global health, despite significant advancements in early LC prevention, diagnosis, and treatment. Within the next 40 years, it is anticipated to overtake IHD as the leading cause of death. The GLOBOCAN-2020 revision on epidemiologic trends of different malignancies agrees with WHO mortality projections. LC’s increased prevalence, incidence, and mortality rates highlight the need for efficient intervention and therapy approaches. The TCGA-c Bioportal data on the number of LC patients who have undergone radiation therapy, targeted therapy, and immunotherapy is 19.2%, 58.9%, and 28.6%, respectively, where cases undergone chemotherapy outnumber the data that do not undergo therapy. The data availability for cases with various therapy to that of ’data does not available’ was 13.2%, 12.1%, 12.1%, and 12% for radiotherapy, chemotherapy, targeted therapy, and immunotherapy, respectively, which raises concerns about the validity of the data.

The use of methods like combination therapy and the exploration of predictive treatment biomarkers and prognosis have resulted from clinical trials of immunotherapy. The therapeutic tactics employed to take advantage of the immunosuppressive properties of TME in the present immunotherapy clinical trial landscape have created difficulties in treatment, clinical testing, and monitoring across diverse tumors. Immunotherapy as a cancer treatment strategy poses difficulties due to the immunosuppression brought on by TME. Microbiome-targeted cancer immunotherapy and the combination of genetic biomarkers with immune-related indicators for more individualized treatment are the new immunotherapy-based strategies. Immune organization in TME is a potent predictor of response to immunotherapy. The basic pathomorphological diagnosis of NSCLC patients could be supplemented with a quick immunological study of the immune response already present in the malignant tissue, such as the presence of lymphocytes and CD8+ macrophages, including PDL1, CXCR3, CCLA4, JAK, TGFβ, and IFNγ expressions, and their placement inside the tumor. The complexity of immune system-tumor communication is complicated, and our understanding of key molecular mechanisms helps us to provide a suitable therapeutic strategy and predict the course of the disease.

The FDA has approved a few combination medicines to boost the clinical effectiveness of ICIs. Combinations of ACT, newer ICIs, cancer vaccines, and small molecule inhibitors are anticipated as action-driven reliable biomarkers for clinical immuno-oncology decision-making well-understood. In this sense, a truly patient-centered, tailored approach is what cancer immunotherapy needs to succeed in the future. Trials on combination therapy using chemotherapeutic drugs with PD1 and CTLA4 checkpoint inhibitors are in clinical trials now. Vaccinations in LC have limited clinical trial evidence to demonstrate definite therapeutic advantages. Observable immune responses may not always correspond to clinically significant responses. While many trials concentrate on patients with distant-stage cancers, patients with stage I or stage II LC with potential recurrence after surgery may be the leading candidates for LC vaccinations. The set of best tumor antigens to target and tackle the different tumor escape mechanisms and many epitopes of a broad set of genes are to be traced for an effective immunotherapy-based LC treatment. 

The multidisciplinary approaches of preclinical cancer research, with contributions from the field of stem cell biology, immunology, and developmental biology, help to decipher the interaction between cancer and stroma. Currently, available models can fully recapitulate patient-tumor phenotypes and responses though various cancer models are available for cancer research. The application of multiple approaches, along with an idea of their limitations, can make improvements in the field of cancer therapeutics. 

## Figures and Tables

**Figure 1 vaccines-10-01963-f001:**
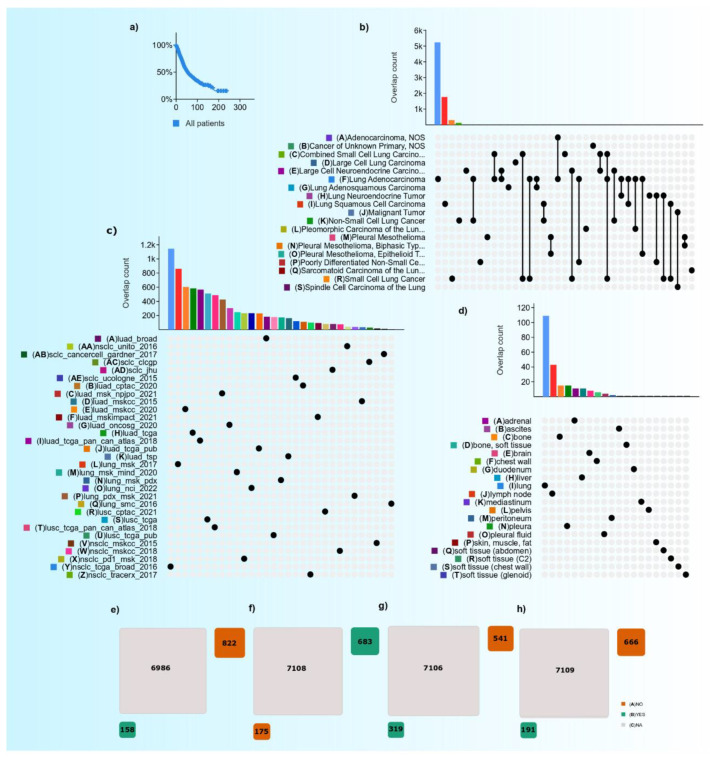
TCGA-cbioportal data on LC. (**a**) KM plot of the LC cases with five-year overall survival: x-axis (% event free), y-axis (time of event in months). (**b**) Frequency of LC histotypes with the overlap in different studies. (**c**) LC projects registered with TCGA. (**d**) PD-L1 tissue site expression in LC cases. (**e**–**h**) Patient records who availed therapies, including radiation therapy, chemotherapy, targeted therapy, and immunotherapy, respectively (Data source: https://www.cbioportal.org (accessed on 1 July 2022)).

**Figure 2 vaccines-10-01963-f002:**
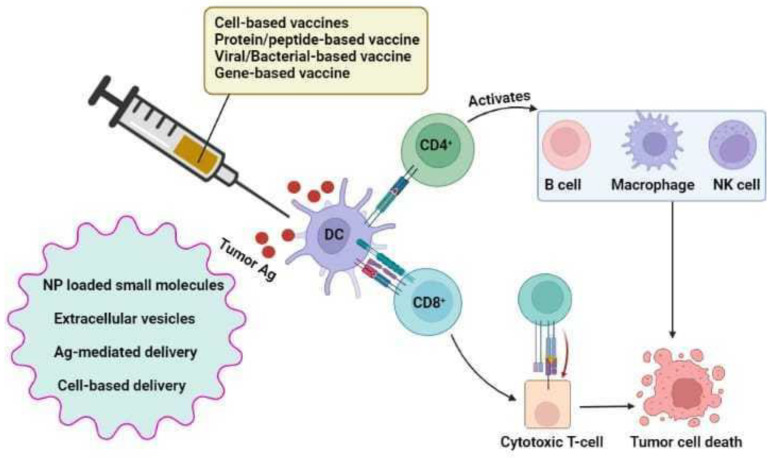
DCs get activated upon vaccine injection and move to the lymph node, wherein antigens are presented to T cells, which are then activated. As CD8+ cells mature because of cytokines released by CD4+ cells, they travel to the cells displaying the target antigen and execute a cytotoxic antitumor response.

**Figure 3 vaccines-10-01963-f003:**
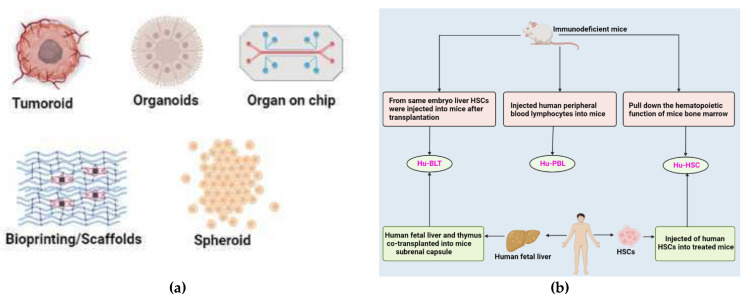
Lung cancer models: (**a)** 3D models currently used to study Lung cancer, cancer stroma, and immunotherapy effects. (**b**) The humanized mouse model of the human immune system involves implanting human hematopoietic cells, lymphocytes, or organs into immunodeficient mice to recreate the human immune system for cancer studies. Hu-BLT, human bone marrow, liver, and thymus; Hu-HSC, human hematopoietic stem cell; Hu-PBL, human peripheral blood lymphocyte.

**Table 1 vaccines-10-01963-t001:** TCGA-cBioportal data comparison on various LC therapy from 31 studies (Data source: https://www.cbioportal.org/ (accessed on 1 July 2022)).

Therapy	No. of Patients Undergone Therapy	No. of Patients Does Not Undergo Therapy	Not Applicable	The Ratio between No. of Patients Undergone to Do Not Undergone (%)	Total Number of Cases with LC Therapy-Data Availability	The Proportion of Subjects with Data Availability and ‘Not Applicable’ (%)
Radiotherapy	158	822	6986	19.2	920	13.2%
Chemotherapy	683	175	7108	_	858	12.1%
Targeted therapy	319	541	7106	58.9	860	12.1%
Immunotherapy	191	666	7109	28.6	857	12.0%

**Table 2 vaccines-10-01963-t002:** FDA-approved drugs used in chemotherapy, targetable therapies, and immunotherapy against major oncogenes in LC (Gene names are italicized).

Therapy	Drugs	Target	Estimated Frequency of Mutation in LC (%)	References
Chemotherapy	Carboplatin, cisplatin, docetaxel, etoposide, gemcitabine, nab-paclitaxel, paclitaxel, pemetrexed, vinorelbine			[19]
Targeted therapy	Afatinib, dacomitinib, entrectinib, erlotinib, gefitinib, osimertinib	EGFR (receptor protein)	15	[20,21,22,23]
	Amivantamab, mobocertinib	*EGFR* (exon 20 insertion)	15	[20,21,22,23]
	Fam-trastuzumab deruxtecan-nxki	HER2	2	[21,24,25]
	Alectinib, brigatinib, ceritinib, crizotinib, loralitinib	*ALK*	5	[20,21,26,27,28,29]
	Ceritinib, crizotinib, entrectinib	*ROS1*	2	[21,30,31,32,33]
	Sotorasib	*KRAS* G12C	25–33	[20,21,25]
	Larotrectinib	*NTRK*		
	Dabrafenib, trametinib	*BRAF* V600E	2	[20,21,25]
	Capmatinib, tepotinib	*MET* (exon 14 skipping)	3	[21,34,35]
	Pralsetinib, selpercatinib	*RET*	2	[20,21,36,37]
Immunotherapy	Atezolizumab, durvalumab, cemiplimab, nivolumab, pembrolizumab	PD1/PDL1 pathway	33	[21,38,39,40,41,42,43]
	Ipilimumab	CTLA4 pathway		[25,44]

The italic represent the gene.

**Table 3 vaccines-10-01963-t003:** Current and completed clinical studies of PD-1 and CTLA-4 ICIs (Data source: https://clinicaltrials.gov/ (accessed on 1 July 2022)) [132].

Agent	Phase	Study Population	Design and Description	Primary Endpoint	Enrolment	NCT
Durvalumab	II	Advanced NSCLC	Evaluating efficacy and safety of the PD-L1 inhibitor durvalumab as first-line therapy	OS	50	NCT02879617
Niraparib	II	NSCLC	Niraparib + Pembrolizumab Niraparib alone Niraparib + Dostarlimab	ORR	53	NCT03308942
Ipilimumab	III	Stage IV/Recurrent NSCLC	Ipilimumab + Paclitaxel/Carboplatin Placebo + Paclitaxel/Carboplatin	OS	1289	NCT01285609
Pembrolizumab	II	Advanced NSCLC	Pembrolizumab + Physician’s choice chemotherapy	PFS	35	NCT03083808
AK104	II	Advanced NSCLC	AK104 +Docetaxel	ORR	40	NCT05215067
Pembrolizumab	II	Metastatic NSCLC	Pembrolizumab + chemotherapy vs. Placebo + chemotherapy	PFS	98	NCT03656094
AK105	III	Metastatic Nonsquamous NSCLC Stage IV	AK105 + Carboplatin and Pemetrexed vs. Placebo + Carboplatin and Pemetrexed	PFS	360	NCT03866980
ONC-392	I & II	advanced or metastatic solid tumors and NSCLC	ONC-392 Treatment as a single agent vs. ONC-392 in combination with pembrolizumab	DLT, MTD, RP2D, TRAE	468	NCT04140526
mRNA Vaccine	I & II	Metastatic NSCLC	BI 1361849 mRNA Vaccine + durvalumab BI 1361849 mRNA Vaccine + durvalumab + tremelimumab	TEAE	61	NCT03164772
KN046	II	Advanced NSCLC	KN046 + Axitinib	ORR	54	NCT05420220

## Data Availability

Not applicable.

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
