# Peer review of "Advances in the Lung Cancer Immunotherapy Approaches"

_vaccines, 2022, doi:10.3390/vaccines10111963_

Round 1

Reviewer 1 Report

The article by Hafiza Padinharayil et al. entitledAdvances in the Lung Cancer Immunotherapy Approaches” is quite interesting and however, suggests fixing the following issue.

1. Despite being a review, the author must write the aim, objectives, and conclusion of the study in the abstract. Authors must write the abstract according to the author's instructions.

2. Authors from non-English-speaking countries should ensure that they have their articles corrected by a native English speaker for grammatical, stylistic, and typographical errors.

3. In title 6. In clinical and preclinical studies of lung cancer, the authors focused mainly on clinical studies rather than preclinical evidence. Authors can emphasize in vitro and rodent studies.

4. Authors can create a table listing immunotherapy and experimental models using different titles in vitro, in vivo, and clinical research. This may make it easier for readers to understand

Author Response

  1. Despite being a review, the author must write the aim, objectives, and conclusion of the study in the abstract. Authors must write the abstract according to the author's instructions.

Response: The aim, background, objective and conclusion is updated in the abstract

  1. Authors from non-English-speaking countries should ensure that they have their articles corrected by a native English speaker for grammatical, stylistic, and typographical errors.

Response: Corrected

  1. In title 6. In clinical and preclinical studies of lung cancer, the authors focused mainly on clinical studies rather than preclinical evidence. Authors can emphasize in vitro and rodent studies.

Response: We have included clinical studies in subhead 6, and preclinical studies considering in vitro and in vivo models in subhead 9.

  1. Authors can create a table listing immunotherapy and experimental models using different titles in vitro, in vivo, and clinical research. This may make it easier for readers to understand

Response: Table 3 belong to immunotherapy approaches and added tables 4 and 5 as per the suggestions that enlisted clinical applications of experimental models.

Reviewer 2 Report

This is an extremely long review article, which reads like a textbook chapter. In a review article, the authors' should focus one or two challenges in the field and try provide evidence based logical analysis with futuristic vision.

The article as it currently stands is extremely superficial and diffuse.  The authors should cut short the text by atleast 50%. Further, they should identify challenges in the field; it could be immunotherapy-personalization, CAR-T cell challenges, vaccines or anthingelse, and go into the depth of analysis.

Author Response

  1. This is an extremely long review article, which reads like a textbook chapter. In a review article, the authors' should focus one or two challenges in the field and try provide evidence based logical analysis with futuristic vision.

The article as it currently stands is extremely superficial and diffuse.  The authors should cut short the text by at least 50%. Further, they should identify challenges in the field; it could be immunotherapy-personalization, CAR-T cell challenges, vaccines or anthingelse, and go into the depth of analysis

Response: We have cross-checked the grammatical error and updated the presentation style of the manuscript. The word count has been reduced to 12,000 from 20,000. We also made significant revisions to make the manuscript direction-based, and to focus on advancements and challenges in immunotherapy.

Round 2

Reviewer 1 Report

Accept in present form

Reviewer 2 Report

No further comments